# Impacts of Casting Scales and Harsh Conditions on the Thermal, Acoustic, and Mechanical Properties of Indoor Acoustic Panels Made with Fiber-Reinforced Alkali-Activated Slag Foam Concretes

**DOI:** 10.3390/ma12050825

**Published:** 2019-03-11

**Authors:** Mastali Mohammad, Kinnunen Paivo, Karhu Marjaana, Abdollahnejad Zahra, Korat Lidija, Ducman Vilma, Alzaza Ahmad, Illikainen Mirja

**Affiliations:** 1Fiber and Particle Engineering, Faculty of Technology, University of Oulu, P.O. Box 4300, 90014 Oulu, Finland; Paivo.Kinnunen@oulu.fi (K.P.); Zahra.abdollahnejad@oulu.fi (A.Z.); Ahmad.Alzaza@student.oulu.fi (A.A.); Mirja.Illikainen@oulu.fi (I.M.); 2VTT Technical Research Center of Finland Ltd, P.O. Box 1300, 33101 Tampere, Finland; Marjaana.Karhu@vtt.fi; 3National Building and Civil Engineering Institute, Dimičeva ulica 12, SI-1000 Ljubljana, Slovenia; lidija.korat@zag.si (K.L.); vilma.ducman@zag.si (D.V.)

**Keywords:** mechanical properties, acoustic properties, thermal insulation properties, lightweight acoustic panels, blast furnace slag, tunnel kiln slag

## Abstract

This paper presents experimental results regarding the efficiency of using acoustic panels made with fiber-reinforced alkali-activated slag foam concrete containing lightweight recycled aggregates produced by using Petrit-T (tunnel kiln slag). In the first stage, 72 acoustic panels with dimension 500 × 500 × 35 mm were cast and prepared. The mechanical properties of the panels were then assessed in terms of their compressive and flexural strengths. Moreover, the durability properties of acoustic panels were studied using harsh conditions (freeze/thaw and carbonation tests). The efficiency of the lightweight panels was also assessed in terms of thermal properties. In the second stage, 50 acoustic panels were used to cover the floor area in a reverberation room. The acoustic absorption in diffuse field conditions was measured, and the interrupted random noise source method was used to record the sound pressure decay rate over time. Moreover, the acoustic properties of the panels were separately assessed by impedance tubes and airflow resistivity measurements. The recorded results from these two sound absorption evaluations were compared. Additionally, a comparative study was presented on the results of impedance tube measurements to compare the influence of casting volumes (large and small scales) on the sound absorption of the acoustic panels. In the last stage, a comparative study was implemented to clarify the effects of harsh conditions on the sound absorption of the acoustic panels. The results showed that casting scale had great impacts on the mechanical and physical properties. Additionally, it was revealed that harsh conditions improved the sound properties of acoustic panels due to their effects on the porous structure of materials.

## 1. Introduction

There is a growing need for more sustainable construction materials. As the world population grows exponentially, construction consumes more and more raw materials. Excessive consumption of raw materials leads to increased energy consumption, emission of greenhouse gases, and dust pollution [1]. One of the main sources of CO_2_ is the production of ordinary Portland cement (OPC), which accounts for up to 5% of total CO_2_ emissions [2,3,4,5]. Alkali-activated materials were proposed and investigated as one alternative to cement-based compositions, because of acceptable mechanical and durability properties in addition to lower negative environmental impacts. By alkali activation, different side streams could be converted and used as sustainable construction materials. The sustainable concrete industry is one research area that is broadly investigated; researchers specifically focus on how to convert industrial by-products into cement and concrete [6,7,8,9].

Recently, these alternative binders received a lot of attention regarding their use as foam concretes, which include the advantages of both porous concretes and alkali-activated binders [10,11,12]. On the other hand, alkali-activated foam binders present eco-friendly and cost-effective construction materials that are lightweight with good thermal and acoustic properties. Different alkali-activated foam binders were developed using fly ash, metakaolin, and granulated blast-furnace slag. To produce foam concretes, different approaches are employed to introduce air voids in alkali-activated binders, such as adding chemical agents (aluminum (Al), hydrogen peroxide (H_2_O_2_), sodium perborate (NaBO_3_·nH_2_O)) and premade foaming agents [10,13]. Using chemical agents, such as aluminum powder, has some disadvantages, including high kinetic energy, which limits the usage of this chemical agent for alkali-activated slag binders because they have short setting times [13,14]. Moreover, this approach generates disorderly large voids that negatively affect the thermal and mechanical properties; however, using this powder has a high impact on density reduction. The mechanical foaming method generates a consistent distribution of fine voids [14], but the density of porous composition cannot always be extensively reduced compared with the chemical foaming agents. Different efforts were made by many researchers on developing alkali-activated foam concretes in lab-scale testing using mechanical foaming methods. Table 1 reports the summarized properties recorded in their investigations.

Although no experimental data are available regarding the use of chemical agents to produce foam concrete in upscaling applications, the premade foaming approach is widely and successfully used to generate OPC-based foam concretes.

To date, all studies regarding the development of alkali-activated foam binders were limited to laboratory scales, and no one developed these porous materials for upscaling applications. However, it should be mentioned that various insulation panels made with the mineral fibers were developed for upscaling applications, and these insulation panels are used in buildings [15].

For the first time, the present investigation represents the impacts of casting scales and harsh conditions on the thermal, acoustic, and mechanical properties of indoor acoustic panels made with fiber-reinforced alkali-activated slag foam concretes. Therefore, there were many unclear issues in this field, which should be responded in different aims.

The main objectives and the research significances of this study were as follows:Assessing the feasibility of using normal OPC-based concrete production lines to produce these lightweight and porous alternative binders;A comparative study on the differences and difficulties between lab-scale and large-scale (or upscaling) foam concrete preparations and castings;Measuring the hardened state (the flexural and compressive strengths), thermal insulation (thermal conductivity, thermal diffusion, and volumetric heat capacity), and sound properties (in a reverberation room and impedance tube) of acoustic panels with dimension 500 × 500 × 35 mm;The effects of lab-scale casting and large-scale casting on the hardened and acoustic properties;Evaluation of hardened and acoustic properties of acoustic panels exposed to different harsh conditions, including freeze/thaw conditions and CO_2_ gas flow exposure.

## 2. Experimental Program

### 2.1. Materials and Mix Design

Fiber-reinforced alkali-activated slag foam concrete was composed of ground granulated blast-furnace slag (GGBFS), recycled aggregates produced by granulation of Petrit-T (tunnel kiln slag), alkali activator solution (a combination of sodium silicate and sodium hydroxide), polyvinyl alcohol (PVA) fiber, and premade foam. The Petrit-T was supplied by Höganäs (Ab, Höganäs, Sweden), which is a stable by-product obtained in the production of sponge iron. More details regarding Petrit-T can be found in References [25,26]. The GGBFS was supplied by Finnsementti (Finnsementti Oy, Espoo, Finland) with d_50_ value and density of the GGBFS as 10.8 μm and 2.93 g/cm^3^, respectively. The chemical compositions of GGBFS were mainly composed of CaO ≈ 39%, SiO_2_ ≈ 33%, Al_2_O_3_ ≈ 9.5%, and MgO ≈ 10%. GGBFS was activated by a combination of sodium silicate and sodium hydroxide as the alkali activator with a ratio of 2. The NaOH solution was prepared by combining NaOH pellets in water with a molar concentration of 12 mol/L and, subsequently, cooling the solution to room temperature. A sodium silicate solution was used as a liquid sodium silicate with a modulus of 2.5 (molar ratio M_s_ = SiO_2_/Na_2_O). The alkali activator-to-binder ratio was equal to 0.56. Moreover, to minimize the negative effects of drying shrinkage, PVA fibers (2.4% of slag mass) were employed to reinforce the foam concrete as one of the effective and easiest solutions [25]. These fibers (RSC 8 mm) were supplied by Kuraray (Okayama, Japan). The PVA fibers had the following physical and mechanical properties: length-to-diameter ratio of 200, elastic modulus of 41 GPa, tensile strength of 1600 MPa, elongation at break of 6%, and density of 1.3 g/cm^3^. A premade foam was used to introduce the air voids. Stirring the foaming agent in water led to reach a stiff white foam with an increase of 20–25 times in volume. The foaming agent, in the form of protein hydrolysate, was supplied by EABASSOC (Manchester, UK). The agent was mixed with water using a weight ratio of 1:33. The created foam was stiff, resembling a shaving foam, with a density of approximately 0.045 g/cm^3^. The diameter of 75–85% of the bubbles was in the range of 0.3–1.5 mm. An EABASSOC junior foam generator was used to produce premade foam for the large-scale casting. This machine could have a throughput of typically 150–200 L of foam per minute, depending on the air compressor used. To have the best foam quality, the foam generator was connected to an air pressure machine with maximum pressure capacity of 10 bar. Recycled aggregates were made by mixing Petrit-T as pre-wetted, borax, and sodium silicate. Pre-wetting was executed by stirring 10% mass of Petrit-T water with Petrit-T and then covering with plastic for 24 h. Using Petrit-T without the pre-wetting process is impossible because it results in the production of a huge amount of heat when the alkali solution is added. The proportions of the mix composition are listed in Table 2.

The small-scale recycled aggregate production procedure was addressed in Reference [25]. This recipe (Petrit-T (100% mass), the ratio of alkali liquid (combination of sodium silicate and borax) to Petrit-T 0.7, and water for pre-wetting of Petrit-T (10% Petrit-T mass)) was also used for producing recycled aggregates in the upscaling process; however, it did not work, and Petrit-T was dry after adding the alkali liquid (a combination of sodium silicate and borax was around 70% mass of Petrit-T). Therefore, a trial-and-error process was used to achieve the minimum content of alkali liquid to consider both economic and cost efficiencies in the up-scaled process. According to the results of this procedure, the content of alkali activator to Petrit-T in upscaling increased from 0.7, which worked in the laboratory scale, to 1. Moreover, as used in the small scale, Petrit-T was pre-wetted using water (10% Petrit-T mass) 24 h before adding the alkali liquid.

In addition, the particle sizes of recycled aggregates were observed to vary in higher ranges of 0.2–30 mm, which may be due to differences in heat generation during the mixing of materials, differences in speed and frequency of the mixer, and differences in the volume of materials. After recycled aggregates were produced using the granulation process, the aggregates were covered with plastic for 24 h. Using coarse recycled aggregates affects the hardened state properties and density of the mix composition; therefore, recycled aggregates were sieved to separate coarse aggregates with diameters larger than 10 mm (see Figure 1). The results obtained from small-scale casting of fiber-reinforced alkali-activated slag foam concrete in the lab conditions met the requirement criteria for using this material as the acoustic panels. Thus, all efforts were made to have similar results in the upscaling pilot as compared to the results obtained in the lab.

### 2.2. Casting and Curing

The mixer used for industrial upscaling had a maximum capacity of 150 L. In the batching process, the dry ingredients, including recycled aggregates and slag, were initially stirred for 1 min (Figure 2a). Then, alkali solution was added, and materials were mixed for an extra 2 min (Figure 2b). PVA fibers were then gradually introduced to the mix composition while stirring the materials to avoid balling effects (Figure 2c). Finally, premade foam was added to the mix composition, and all materials were combined for a further 1 min (Figure 2d). The fresh fiber-reinforced alkali-activated slag foam concrete was moved by truck to a place where the oiled molds had been prepared. It is worth stating that, to transfer fresh concrete by truck, the fresh concrete must have good workability.

As indicated in Figure 2e,f, fresh foam concrete was cast into the oiled molds, and the surfaces of the panels were flatted and then covered with plastic bags. After 48 h, the specimens were demolded and then again wrapped with plastic for 28 days (Figure 2g,h).

### 2.3. Test Procedure

#### 2.3.1. Compressive and Flexural Strengths

Mechanical properties were investigated based on the compressive and flexural strengths. Because the fiber orientation and dispersion affect the flexural performance of fiber-reinforced composites, some prismatic beams with dimensions 350 × 100 × 35 mm were extracted in both transverse and longitudinal directions (see Figure 3a). The prismatic beams were assessed under a three-point bending test using a displacement control and a speed rate of 0.6 mm/min. Moreover, the cubic specimens with an edge of 35 mm were cored from the panels, and compressive loading was imposed. The applied load with a speed rate of 1.8 mm/min was submitted in two different directions (F_1_ and F_2_) separately to assess the effects of fiber dispersion on the compressive performance of fiber-reinforced alkali-activated slag foam concretes (see Figure 3b).

#### 2.3.2. X-Ray Micro-Computed Tomography (CT) Technique

X-ray micro-computed tomography (CT) using an XRadia μCT-400 tomograph (XRadia, Concord, CA, USA) was used to investigate the structural characteristics of the specimens. The beam energy and the intensity were set to 140 kV and 70 µA, respectively. With this method, 1600 projection images at an exposure time of 2–3 s per projection were acquired on the charge-coupled device (CCD) camera. Avizo Fire three-dimensional (3D)-image analysis software (Thermo Scientific™ Avizo™ Software, Thermo Fisher Scientific, Waltham, MA, USA) was used to reconstruct the 3D internal pore structure of the samples, as well as to estimate the overall porosity and pore size distribution [27]. The X-ray CT scanning of specimens was conducted to evaluate the porosity with air pores, where the resolution of scanned samples was 33 µm for one pixel. However, all samples fit within the field of view in the detector’s horizontal plane. For porosity determination, a median filter was used. Moreover, segmentation was performed via histograms.

#### 2.3.3. Carbonation Test

As indicated in Figure 4a, five panels were placed in the carbonation chamber for seven days with a pressure of 1 bar, temperature of 23 °C, and relative humidity of 35%. Using this test, the absorbed mass and the carbonated level were assessed. CO_2_ reacts in the foam concrete pores with calcium sources and forms crystals (mainly calcite and sodium carbonate). Therefore, it was assumed that the mass of the panels should increase due to the absorption of carbon dioxide within their highly porous structure. As an indicator solution, phenolphthalein was applied to the surfaces of the panels. If the area remained colorless, this suggested carbonation, and if a panel’s surface color changed to purple, this indicated a high pH value (above 8.6). Because the panels absorb carbon dioxide during their service life, they were used to examine the impacts of carbonation on flexural and acoustic properties.

#### 2.3.4. Freeze/Thaw Test

To execute this evaluation, the temperatures were varied from −18 °C to +20 °C according to the CEN/TS 12390-9:2006 standard [28]. Regarding this standard, the specimens were kept for 13 h at −18 °C and for 3 h at +20 °C (see Figure 4b). The transitions from positive to negative and negative to positive temperatures took 3 h and 5 h, respectively. Each cycle was 24 h in duration. This assessment was carried out using 50 cycles. In this assessment, the panels did not contact directly with water because the panels were used as indoor applications and were never exposed to water. However, panels do have a high possibility of being exposed to high relative humidity; therefore, a container with water was placed beside the panels to increase the humidity. After implementing this assessment, the panels’ mass and the effects of the freeze/thaw test on the flexural and acoustic panels were also investigated. On the other hand, the panels’ performances were evaluated under freeze/thaw conditions exposed to high relative humidity.

#### 2.3.5. Impedance Tube and Airflow Resistivity

The sound absorption of fiber-reinforced alkali-activated foam concrete used in acoustic panels was conducted by employing a 34.8-mm impedance tube. When a propagating sound wave impinges on a surface, a portion of the sound energy is absorbed, and the rest is reflected. The sound absorption coefficient (*α*) determines the absorbed sound energy from the total incident energy. This coefficient was determined experimentally by adhering to ISO 10534-2 [29]. Figure 5a shows the apparatus used to determine the absorption coefficient. The specimens were placed in the sample holder, shown on the right-hand side of Figure 5a, with the piston adjusted for zero air gap behind the sample. Two microphones were installed along the axis of the tube to measure the sound pressure. The absorption coefficient was computed using
*α* = 1 − |*R*|^2^,(1)where *R* is the reflection factor, which can be calculated using
(2)$$R=\frac{H_{12}\;-\;e^{-jk_{0}\left(d_{2}-d_{1}\right)}}{e^{jk_{0}\left(d_{2}-d_{1}\right)}\;-\;H_{12}}e^{2jk_{0}d_{2}},$$where *j* is $\sqrt{-1}$, *k*_0_ is the wave number, *H*_12_ is the transfer function estimate between the microphones, and *d*_1_ and *d*_2_ are the distances from the flat surface of the specimen to the near and far microphones, respectively. In this study, the frequency range was varied in the range of 250–5000 Hz. The sound absorption coefficient was computed based on averaging the results of five samples.

The airflow resistivity indicates the ability of sound absorption in a material. A continuous airflow method was used to evaluate the airflow resistivity, wherein air was inserted inside the specimens with a specific pressure, and the pressure loss was obtained.

The airflow resistivity was influenced by different parameters such as the pore size, distribution, and shape, and the tortuosity of the air voids. An optimal flow resistivity for sound absorption maximization could be achieved via the interconnected pore structure in combination with the sufficient air void content.

Figure 5b shows the apparatus used to measure the airflow resistivity with circular specimens. The circular exposed area of the material sample was 6400 mm^2^ (diameter of the exposed area was 90 mm). Airflow through the material was measured with a laminar flow element (20 L/min), and the pressure drop was measured with a dpm TT 570 micromanometer (DP Measurement, Buckingham, UK). The flow velocity through the materials was 2.5, 5, 10, and 20 mm/s. Three parallel measurements for each sample were made, as shown in Figure 5b. The flow resistivity (*R*) was calculated using
(3)$$R=\frac{A\cdot \Delta P}{d_{s}\cdot Q},$$where *A* is the area of the sample (m^2^), Δ*P* is the pressure drop of the sample (Pa), *d_s_* is the thickness of the sample (m), and *Q* is the airflow rate (m^3^/s). In this study, the airflow resistivity was obtained by varying the airflow rates as follows: *q* = 1.6 × 10^−5^, 3.2 × 10^−5^, 6.4 × 10^−5^, and 1.3 × 10^−4^ m^3^/s.

#### 2.3.6. Measurement of the Sound Absorption in a Reverberation Room

The panels were installed on the floor whose surface pointed upward onto the reverberation chamber floor (see Figure 6). This assessment was carried out on the premises of VTT Expert Services Ltd., (Espoo, Finland). The volume and the inner surface area of the reverberation room were 201 m^3^ (height of 4.7 m and floor area of 5.95 m × 7.20 m) and 209 m^2^, respectively. Moreover, the temperature and the relative humidity of the reverberation room were 21 °C and 52%, respectively, when measurements were carried out.

The sound absorption coefficient (*α_s_*) and the sound absorption rating (*α_w_*) were determined based on SFS EN ISO 354-2003, [30] and SFS EN ISO 11654-1997 [31], respectively. The airborne sound reduction index of the board was defined using two channel sound pressure level measurements, whereby two sources were fixed and microphones moved.

The measurement equipment used included the condenser microphone (Brüel & Kjær, B&K 4134, Helsinki, Finland), microphone preamplifier (B&K 2660), rotating microphone boom (B&K 3923), power amplifier (Peavey PV 2600, Peavey Electronics Corporation, Mississippi, USA), loudspeakers (Sinmarc V121L, VTT, Espoo, Finland), real-time analyzer (Norsonic 830, Norsonic, Tranby, Norway), and sound calibrator (B&K 4228).

According to the reverberation measurements in the empty reverberation room, the equivalent sound absorption *A*_1_ was computed (per frequency band) based on Equation (4), expressed in m^2^ as
(4)$$A_{1}=\frac{55.3V}{cT_{1}}-4Vm_{1},$$where *V* is the volume of the reverberation room (m^3^), *T*_1_ is time in the empty reverberation room (s), *m*_1_ is the power attenuation coefficient in the empty room, computed based on formula ($m=\frac{\alpha }{10\log e})$ (m^−1^), and *c* is the speed of sound in the air (m/s), calculated according to (*c* = 331 + 0.6*t*). In this equation, *t* is the temperature, and it is valid for temperatures between 15 °C and 30 °C.

The equivalent sound absorption *A*_2_ for the room occupied with the test specimens was calculated according to Equation (5), also indicated in *m*_2_ as
(5)$$A_{2}=\frac{55.3V}{cT_{2}}-4Vm_{2},$$where *c* and *V* have the same definition as in Equation (4), while *T*_2_ is the reverberation time of the reverberation room with the test specimen placed inside (s), and *m*_2_ is the power attenuation coefficient in the room. The equivalent sound absorption (*A)* was calculated based on Equation (6), expressed in m^2^ as
(6)$$A=A_{2}-A_{1}.$$

The sound absorption coefficient (*α_s_*) can be computed according to Equation (7) when a plane of the specimen has an area between 10 and 12 m^2^.
(7)$$\alpha_{s}=\frac{A}{S},$$where *S* is the area of the tested specimen.

Moreover, the ASTM 423 recommendation provided a similar test criterion to EN ISO 354, called the noise reduction coefficient (NRC) [32]. This was computing using
(8)$$NRC=\frac{\alpha_{250Hz}+\alpha_{500Hz}+\alpha_{1000Hz}+\alpha_{2000Hz}}{4},$$where *α* is related to the practical sound absorption coefficient.

#### 2.3.7. Thermal Insulation

The transient plane source (TPS) technique was used by employing a hot disc thermal constant analyzer based on the procedure of a transiently heated plane sensor. The specimens for this evaluation had dimensions of 30 (height) × 70 (length) × 70 (width) mm. The most critical issue for measuring the thermal properties of the specimens was providing a completely smooth and parallel surface; therefore, all samples were polished flat and parallel.

The humidity has a great impact on the values of thermal conductivity [33,34]; however, the samples needed to be dried before assessing. Zhang et al. adopted a short drying period (6 h) at a moderate temperature (80 °C) with respect to the assumption that the weight loss did not exceed 1% [20]. Therefore, this approach was also employed to dry the samples.

## 3. Results and Discussion

### 3.1. Different Effects of Harsh Conditions on Hardened and Sound Properties of the Panels

#### 3.1.1. Effects of Carbonation and Freeze/Thaw Conditions on Density

Figure 7 shows the changes in the density of the exposed panels to unpressurized 5% CO_2_ gas with a pressure of 1 bar after seven days. According to the results, the density for all panels increased on average more than 6% due to penetration of CO_2_ gas into the porous structure of the panels, consumption of the calcium and sodium sources, and forming of the crystals. During the service life of the panels, CO_2_ was absorbed within the panels, affecting their mechanical and acoustic properties.

The density variations of the panels in freeze/thaw conditions were measured as shown in Figure 8. As indicated, freeze/thaw conditions led to a very slight observable increase of the panel’s density (lower than 1%). Although density loss was expected after the freeze/thaw test due to the formation of damages, an increase in the density of the panels was noticed, which could be due to absorbed humidity.

It is worth stating that placing the water container inside the machine increased the internal humidity greatly, which intensified the negative impacts of freeze/thaw conditions on damaging the internal porosity. Increasing the humidity leads to more ice volume formation inside the pores, and this expansion submits the pore walls to compressive loading. On the contrary, melting results in releasing this imposed internal stress. Each cycle of the freeze/thaw test could be simulated as an imposed fatigue load to the pore walls, in which the pore structure was damaged and altered after a certain freeze/thaw cycle. Wang et al. showed that increasing relative humidity in a freeze/thaw test increased the number of larger pores (diameter >100 nm), while that of smaller pores (diameter <100 nm) decreased [35]. This phenomenon affects the sound absorption properties of the panels.

#### 3.1.2. Effects of Carbonation and Freeze/Thaw Conditions on the Compressive and Flexural Strengths

The impacts of different aggressive conditions on the flexural performances of extracted beams were compared in Figure 9. As shown, the flexural performances of beams extracted from longitudinal and transversal directions were different. These differences could be addressed by fiber dispersion. According to the results, exposing the panels to CO_2_ gas resulted in forming the crystals and forming pores. This increased the flexural strength compared to others (30% and 50% increase in flexural strength in the longitudinal and transversal directions, respectively), while the ductility of the panels was reduced. Ductility (*μ*) is defined as a ratio of the corresponding deflection at flexural strength to the maximum deflection in linear behavior [36]. The significant effects of the aggressive conditions on the flexural performance were detected in the panels that experienced carbonation. However, these panels showed ductile behavior with ductility indices of 2.54 and 3.68 in the longitudinal and transversal directions, respectively. The formation of the crystals affects the chemical bond properties between PVA fibers and the surrounding matrix. Moreover, it was found that carbonation had no great impact on the flexural stiffness, while the flexural stiffness decreased for beams that experienced freeze/thaw conditions, and it was extracted from longitudinal and transversal directions. It can be concluded that freeze/thaw conditions damaged the internal pore structure, although the fiber-bridging action of PVA fibers could compensate for the imposed internal damage and provide high ductility (*μ*: 5.6) particularly in the transversal direction. Figure 9c shows the formation of multiple cracks, which justifies the hardening deflection behavior of prismatic beams.

As expected, similar results with flexural performance were also noticed for compressive strength (see Figure 10). The formation of the crystals filled the pores, and this filling increased the compressive strength more than 30% compared to the specimens cured in ambient conditions. As mentioned earlier, the compressive strength for the extracted cubic specimens that experienced freeze/thaw conditions decreased by 20% compared to the cured specimens in ambient conditions due to damages to the internal structure.

#### 3.1.3. Effects of Carbonation and Freeze/Thaw Conditions on the Pore Structure

The air voids and porosity of fiber-reinforced alkali-activated slag foam concrete in planar and 3D scales are shown in Figure 11. The pore structure of the porous material is indicated as an almost open-cell structure. Regarding the results obtained from the X-ray CT scanning technique, total porosity was measured at 56.4%, the bulk volume (including the solid and void components: 2.95 × 10^12^ mm^3^) to solid volume ratio was 1.29× 10^12^, and the volume of air voids to the volume of pores was 1.66× 10^12^. Note that fibers were not observed due to their low contrast and density, and they are included as pore pace in the calculation of porosity.

Optical images were used to indicate the effects of harsh conditions on the porosity of acoustic panels, as indicated in Figure 12. Comparing the images revealed that the submitted aggressive conditions changed the pore size, porosity, and tortuosity of cellular structures. According to the observations, these damages were more significant in the panels exposed to freeze/thaw conditions and then carbonation than those panels in ambient conditions. Under freeze/thaw conditions, the thickness of the cell walls was reduced, and some cells were even merged, which increased the porosity and tortuosity.

#### 3.1.4. Effects of Carbonation and Freeze/Thaw Conditions on the Sound Absorption Coefficient and Airflow Resistivity

The impacts of imposing different harsh conditions on the sound absorption coefficients are shown in Figure 13. Up to 600 Hz, exposing panels under different conditions had no significant influence on the sound absorption. The thickness of the panels majorly affects the sound absorption at low frequencies; therefore, increasing the thickness increases the sound absorption coefficients [20].

Regarding the results, the panels that experienced freeze/thaw conditions exhibited the best performance in terms of sound absorption coefficients for medium- to high-frequency sound (600–5000 Hz). As shown in Figure 12, this is due to the changes in pore size, porosity, and tortuosity introduced though icing and melting. Moreover, except for the amplitude of sound absorption, it was observed that the imposed aggressive conditions had no great impact on the trend of the transmitted sound waves under different wavelengths. The significant differences in the sound properties of the panels that experienced freeze/thaw and carbonation conditions were related to the medium-frequency range of 600–1500 Hz. The tested panels exhibited better acoustic absorption properties in the range of 125–5000 Hz compared to the developed fly-ash-based geopolymer foam concretes in References [20,37,38].

At the lab-scale testing, ceramsite porous concrete indicated an average acoustic absorption coefficient >0.5 [39] at the medium- to high-frequency regions, while the tested panels at upscaling showed an average acoustic absorption coefficient of 0.45 for the panels kept at the ambient conditions. Note that the sound absorption coefficients of acoustic panels developed here were more than 0.55 at the medium- to high-frequency regions at lab-scale (small-scale) casting and testing [25]. Cementitious composites absorb carbon dioxide gas during their service life, and exposing cementitious composites to CO_2_ gas under a carbonation test is an accelerated approach to estimating their performances in the long term. Therefore, as acoustic panels absorb carbon dioxide during their service life, this carbon dioxide absorption enhances the mechanical and acoustic performances of the acoustic panels.

The NRCs were calculated from the absorption spectrum and are illustrated in Figure 14. A higher NRC reflects higher efficiency in the reduction of the sound echoing in a room covered with acoustic panels. According to the results, the maximum and minimum NRC were measured at 0.46 and 0.38 for the panels exposed to freeze/thaw and ambient conditions, respectively.

Interestingly, comparing the NRC values revealed that the scale of casting had no great impact on the efficiency of porous materials in reducing sound echoing, and similar values were reported for both cases.

Regardless of the tested acoustic panels, all panels showed much higher NRC compared to the developed cellular alkali-activated fly ash concrete in References [37,38] with NRC <0.25. Kim et al. assessed the NRC for high-performance concrete, and the NRC varied from 0.15 to 0.49 [41]. However, a comparison was implemented in Figure 14 to evaluate the NRC in the developed acoustic panels with other foamed materials and acoustic products reported in Reference [40]. According to the presented intervals for NRC values for acoustic or foamed materials, the NRC values varied in the range of 0.35–0.5.

Figure 15 shows the resistance of the acoustic panels exposed to different harsh conditions to the different airflow rates. As expected, the size and tortuosity of pores were affected by different conditions and confirmed the observations in Figure 12. On average, the airflow resistivity was reduced by 40% compared to the ambient conditions, meaning that the pores were connected, and the porous matrices were damaged through exposure in aggressive conditions. For lab-scale casting (small-scale casting), the airflow resistivity was assessed using different airflow rates (2.4 × 10^−5^ to 19 × 10^−5^), and an airflow resistivity of 165,000 Nm/s^4^ was measured [25]; for upscale panels, however, this resistivity was measured around 150,000 Nm/s^4^. This confirms the impacts of casting scales on the pore structures and mechanical properties of porous concretes.

### 3.2. The Sound Absorption of Acoustic Panels in a Reverberation Room

The sound absorption coefficient for each third octave band and the practical sound absorption coefficient of the panels at different frequencies are indicated in Figure 16. It is worth stating that the acoustic panels used in this stage were kept at the ambient conditions. Regarding the results, the sound absorption coefficient consistently increased above 1000 Hz and reached 0.55 at the frequency of 5000 Hz. A comparative study was implemented in Figure 16a to show the efficiency of the panels compared to other construction and building materials. At a frequency of 4000 Hz, except the green walls developed in Reference [42] used as the passive acoustic insulation system for buildings, all materials indicated lower sound absorption coefficients, such as glass for windows, marble/tile, wood, inorganic polymeric foams (comprising a mixture of 70% metakaolin and 30% blast-furnace slag), and concrete block. Moreover, it was detected that the developed panels with fiber-reinforced alkali-activated slag foam concrete had a similar sound absorption capacity to green walls with 71% greenery coverage density at a frequency of 5000 Hz, and green walls with full greenery coverage density indicated a lower sound absorption coefficient.

The weighted sound absorption coefficient (*α_w_*) was also computed as equal to 0.4, which could be categorized as sound absorption class D. NRC was also computed as equal to 0.4, while this coefficient for inorganic polymeric foam as a sound-absorbing and insulating material was achieved equal to 0.25 [23].

As indicated in Figure 17, a comparative evaluation was carried out on the sound absorption of fiber-reinforced alkali-activated slag foam binders under two different scales and approaches. Based on the results, the sound absorption provided by impedance tube measurements indicated higher coefficients than the sound properties recorded in the reverberation room at low to medium (up to 1000 Hz) frequency. From the frequency of 1000 to 5000 Hz, the sound properties recorded in the reverberation room showed higher coefficients than the impedance tubes. In general, these two sound evaluations indicated similar sound properties for the acoustic panels. The assessment of the acoustic properties in a reverberation room was more complex than the assessment by impedance tube measurements; therefore, it seems that the impedance tube has good and acceptable accuracy for measuring the acoustic properties of fiber-reinforced alkali-activated slag-based foam concretes.

Regardless of the frequency, the main differences in the sound absorption coefficients were observed in the scale of foam concrete preparation (around 60%), which majorly affected the pore size, porosity, and tortuosity.

### 3.3. Thermal Insulation Properties

The thermal properties of the panels were evaluated in terms of thermal diffusivity, conductivity, and volumetric heat capacity with a temperature of 24 °C and a relative humidity of 32%. Two pairs of specimens were used to obtain the thermal properties. These assessments revealed that thermal conductivity was equal to 0.27 W/m·K, thermal diffusion was 0.195 mm/s^2^, and volumetric heat capacity was 1.41 MJ/m^3^·K. Stolz et al. reported that cellular alkali-activated fly-ash binders with densities of 0.95–1.3 g/cm^3^ had varied thermal conductivity within the interval of 0.225–0.31 W/m·K, thermal diffusion of 0.18–0.2 mm/s^2^, and volumetric heat capacity of 1.2–1.7 MJ/m^3^·K [37]. Zhang et al. indicated that foam fly-ash-based geopolymers with densities of 0.72–1.6 g/cm^3^ could have thermal conductivities of 0.15–0.48 W/m·K and thermal diffusion of 0.28–0.35 mm/s^2^ [20]. Henon et al. showed that potassium geopolymer foams made with silica fume pore-forming agent for thermal insulation have thermal conductivities of 0.125–0.35 W/m·K with densities of 0.4–0.8 g/cm^3^ [43]. For OPC-based foam concretes, the thermal conductivity varied from 0.15 to 0.6 W/m·K with densities of 0.4–1.7 g/cm^3^ [44]. Comparing the results revealed, the developed acoustic panels have acceptable thermal insulation properties with low thermal conductivity and thermal diffusivity.

## 4. Application of Acoustic Panels for Indoor Walls

Despite the foamed concrete generated by the chemical agents, the developed acoustic materials in this study could be designed based on the required volume and could be easily used for casting the desired shapes due to their high flowability. Moreover, the panels could be coated and colored to use as decorative objectives (see Figure 18) and assembled or attached to the wall using bolts and nuts. Drilling would not propagate cracks on the panel surfaces due to reinforcement from using PVA fibers.

## 5. Conclusions

This study presents the experimental results regarding the development of fiber-reinforced alkali-activated slag foam concretes in upscaling. Mechanical, acoustic, and thermal properties of acoustic panels were assessed, resulting in the following conclusions:Scale of granulation to produce lightweight recycled aggregates had significant effects on the size and quality of the final produced aggregates. The maximum influence was noticed in the required liquid to granulate.Normal concrete production lines can be used to produce these lightweight and porous alternative materials.Panels were resistant to freeze/thaw and carbonation conditions. The maximum compressive (2 MPa) and flexural (1.6 MPa) strengths were recorded in the specimens exposed to carbonation, while the best performance in terms of ductility under flexural loading was observed in the specimens that experienced freeze/thaw conditions (more than 5). Moreover, it was found that aggressive conditions had greater impact on the mechanical properties compared to PVA fiber orientation.In low-frequency regions (up to 500 Hz), the aggressive conditions had no influence on the acoustic properties. The best sound performance in terms of sound absorption was found in the panels that had undergone freeze/thaw conditions (with the maximum sound absorption >0.7).The NRCs of acoustic panels varied in the range of 0.35–0.5.The sound absorption measurements in a reverberation room showed that the acoustic panels have similar sound absorption capacity to green walls with 71% greenery coverage density at a frequency of 5000 Hz with a sound absorption coefficient of 0.55, and green walls with full greenery coverage density indicated a lower sound absorption coefficient (0.5).Similar sound absorption capacity was observed by impedance tube measurements and the reverberation room for the acoustic panel. Impedance tube measurements indicated good accuracy for assessing the sound absorption properties, and the use of impedance tube measurements is recommended regarding the complexity of sound absorption properties in a reverberation room.The scale of casting acoustic panels (lab scale or upscaling) indicated great influences on the sound absorption coefficients, such that better performances were noticed in the lab-scale testing (with a sound absorption coefficient of 0.85) compared to upscaling (with a sound absorption coefficient of 0.55).Thermal insulation properties of the acoustic panels with thermal conductivity of 0.27 W/m·K, thermal diffusion of 0.195 mm/s^2^, and volumetric heat capacity of 1.41 MJ/m^3^·K are in the ranges of other alkali-activated foam concretes, and the results exhibit better thermal insulation than OPC-based foam concretes with thermal conductivity of 0.15–0.6 W/m·K.

## Figures and Tables

**Figure 1 materials-12-00825-f001:**
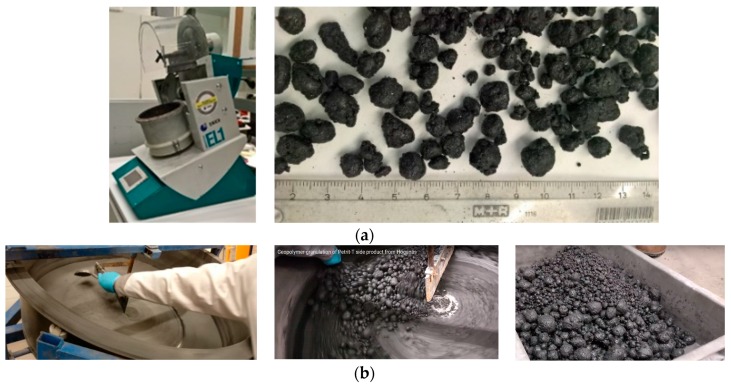
Granulation of Petrit-T to produce recycled aggregates at the (**a**) laboratory scale, and (**b**) pilot scale.

**Figure 2 materials-12-00825-f002:**
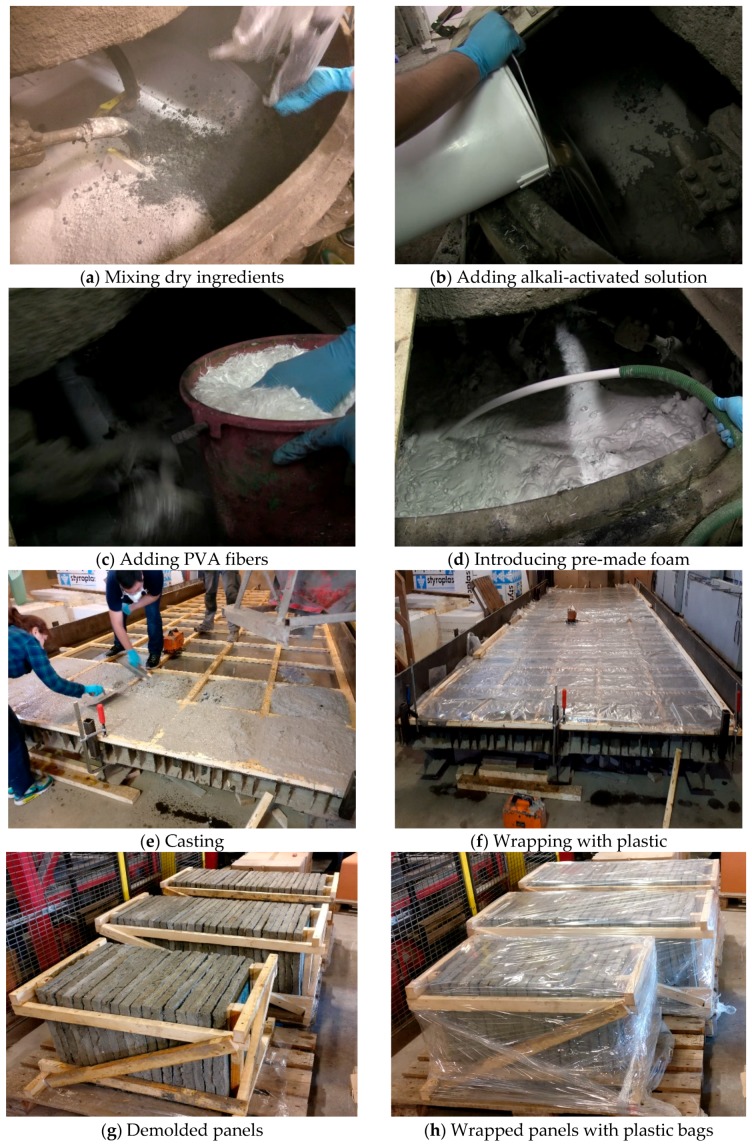
Sequences of mixing, casting, and curing. (**a**) Mixing dry ingredients; (**b**) adding alkali-activated solution; (**c**) adding polyvinyl alcohol (PVA) fibers; (**d**) introducing pre-made foam; (**e**) casting; (**f**) wrapping with plastic; (**g**) demolded panels; (**h**) wrapped panels with plastic bags.

**Figure 3 materials-12-00825-f003:**
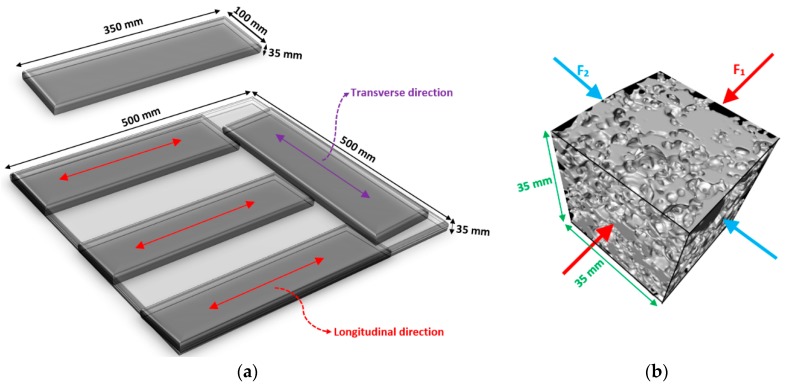
(**a**) The extracted prismatic beams; (**b**) directions of applied compressive loads.

**Figure 4 materials-12-00825-f004:**
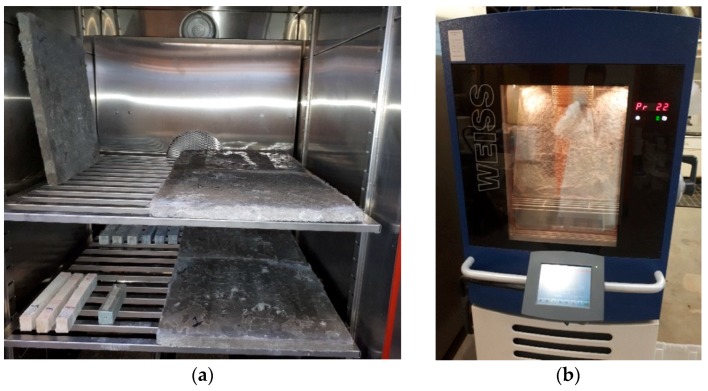
(**a**) Carbonation chamber; (**b**) freeze/thaw test.

**Figure 5 materials-12-00825-f005:**
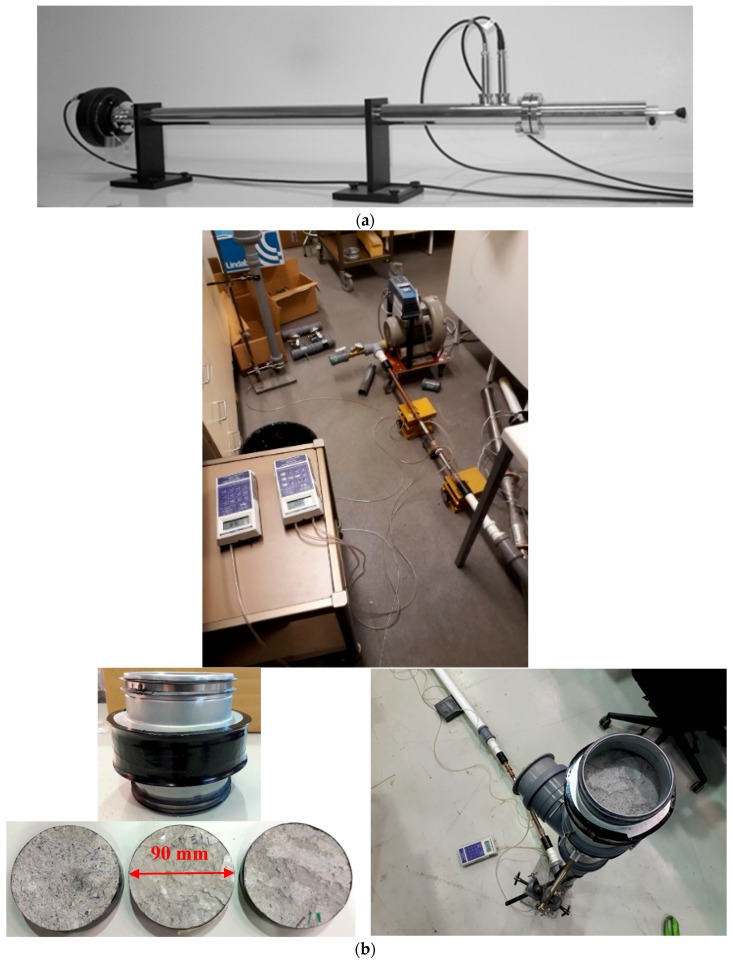
(**a**) Apparatus used to measure the acoustic indicators; (**b**) apparatus employed to measure airflow resistivity [25].

**Figure 6 materials-12-00825-f006:**
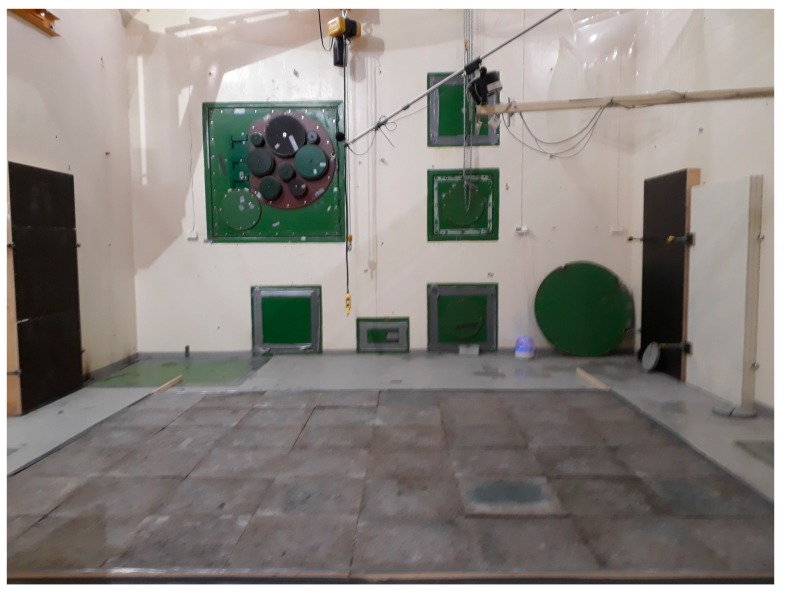
The panels installed onto the reverberation chamber floor.

**Figure 7 materials-12-00825-f007:**
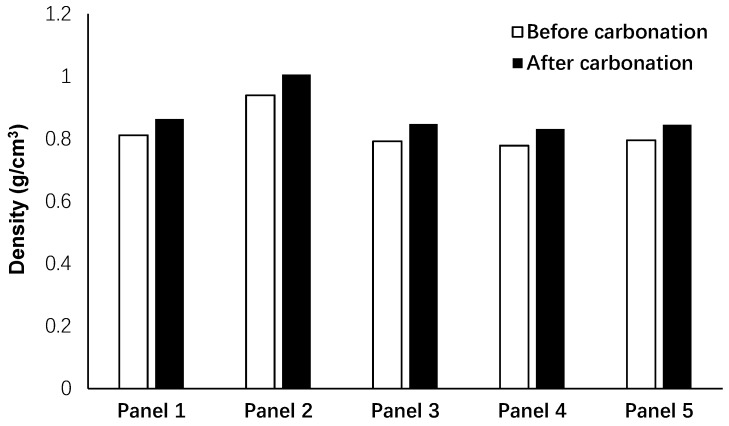
Effects of carbonation on the density of the panels.

**Figure 8 materials-12-00825-f008:**
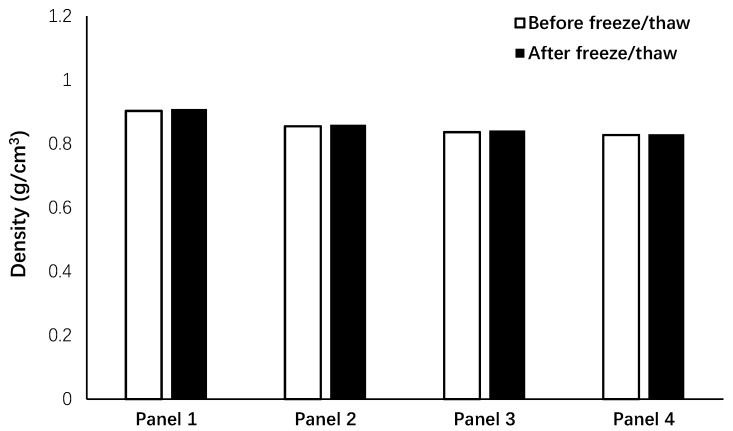
Effects of freeze/thaw conditions on the density of the panels.

**Figure 9 materials-12-00825-f009:**
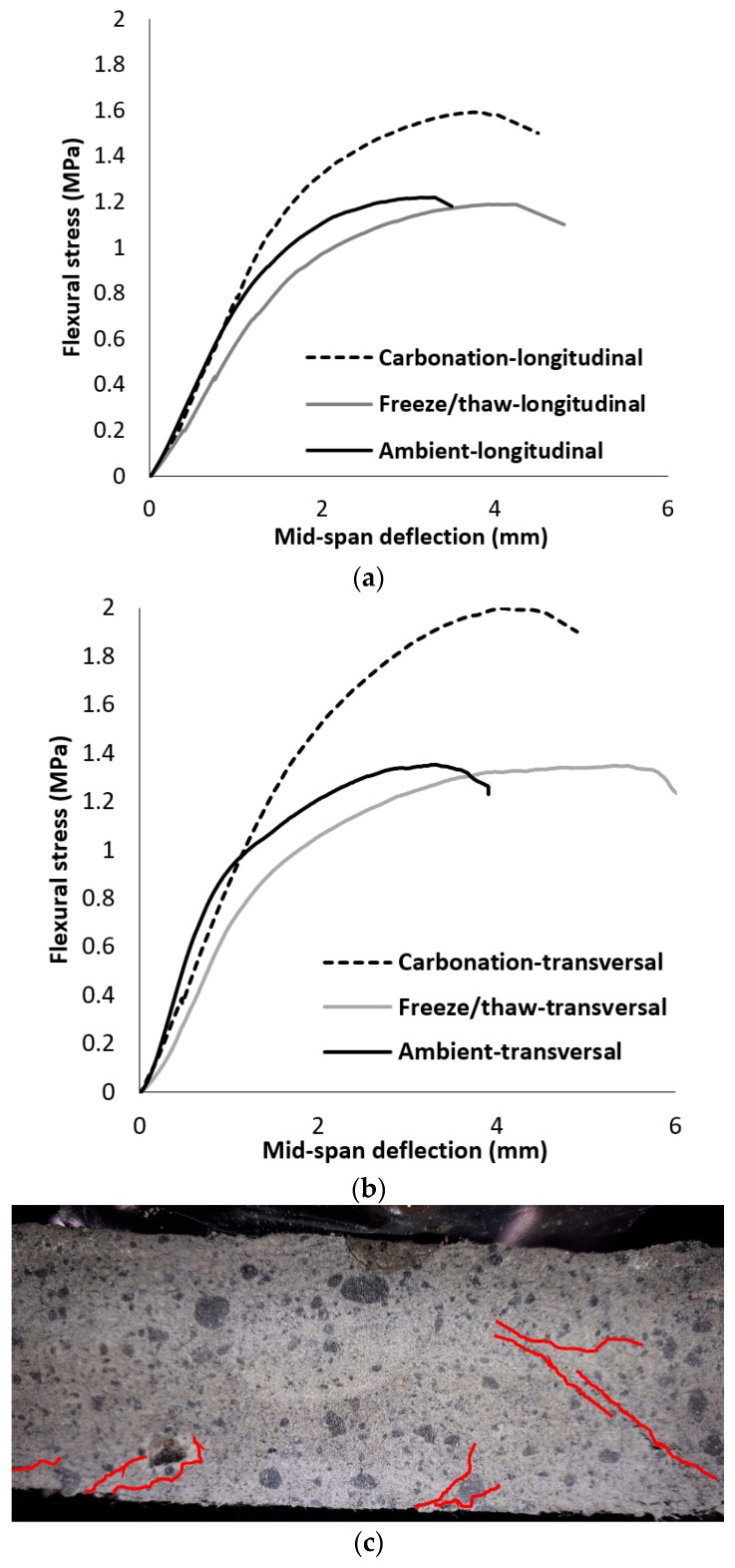
Effects of fiber orientation on the flexural performance of the prismatic beams: (**a**) extracted in the longitudinal direction; (**b**) extracted in the transversal direction; (**c**) crack patterns.

**Figure 10 materials-12-00825-f010:**
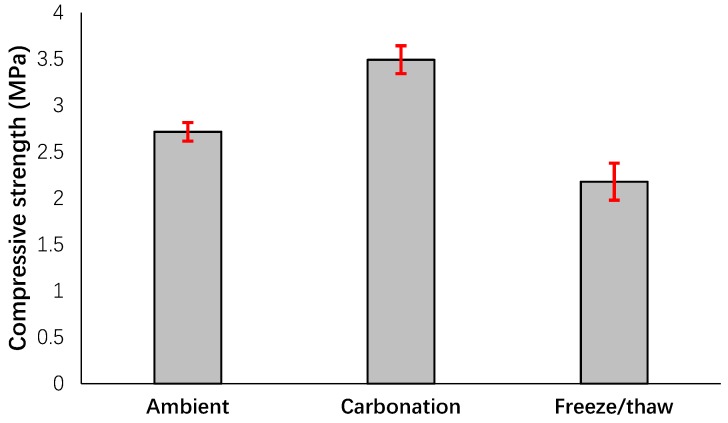
Effects of different conditions on the compressive strength.

**Figure 11 materials-12-00825-f011:**
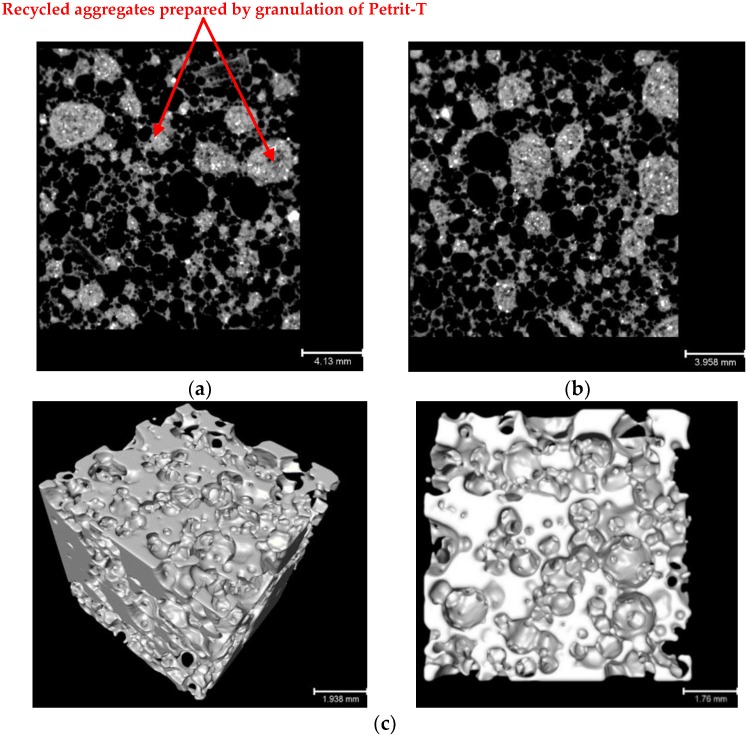
(**a**) Cross-section in *X*–*Z* direction; (**b**) cross-section in *Y*–*Z* direction; (**c**) three-dimensional (3D) segmentation of inner section.

**Figure 12 materials-12-00825-f012:**
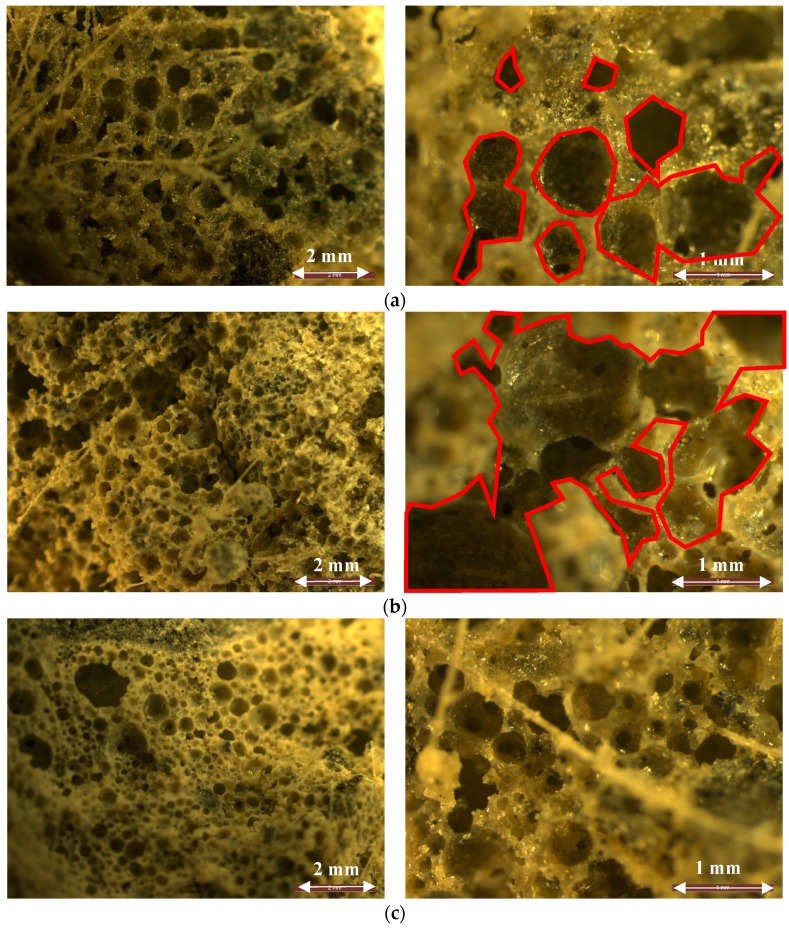
Optical images from the porous matrix exposed to (**a**) ambient, (**b**) freeze/thaw, and (**c**) carbonation conditions.

**Figure 13 materials-12-00825-f013:**
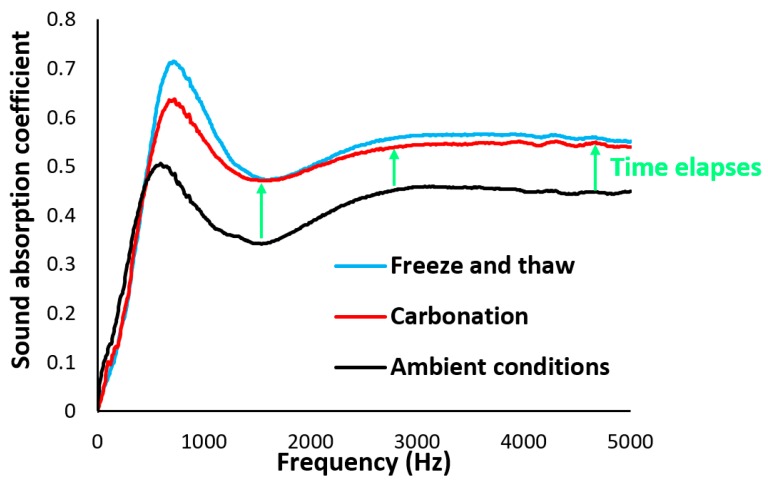
Sound absorption of acoustic panels under different conditions.

**Figure 14 materials-12-00825-f014:**
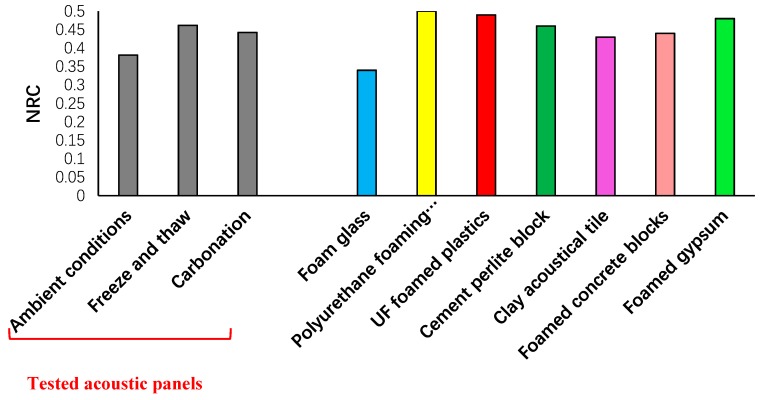
Noise reduction coefficients (NRCs) in the developed acoustic panels and other foamed materials reported in Reference [40].

**Figure 15 materials-12-00825-f015:**
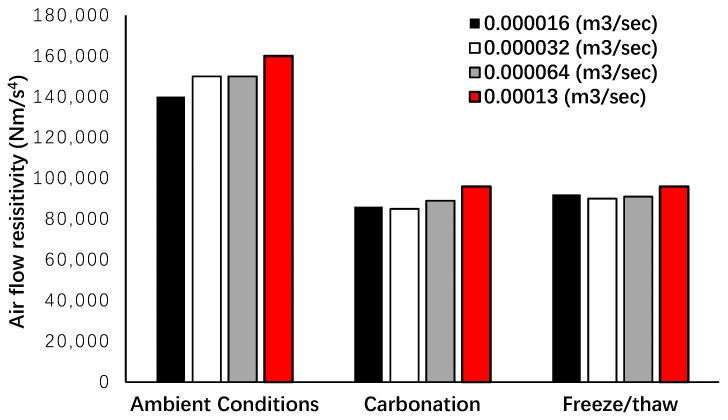
Airflow resistivity of acoustic panels under different conditions.

**Figure 16 materials-12-00825-f016:**
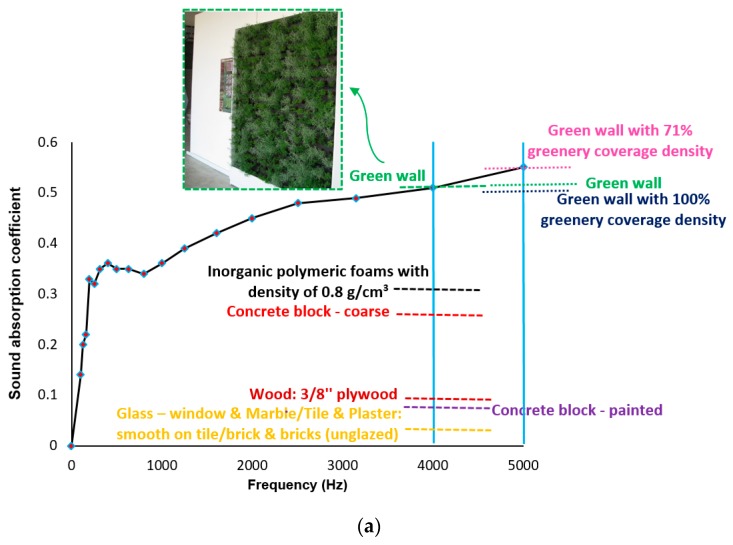
(**a**) The sound absorption coefficient; (**b**) the practical sound absorption coefficient.

**Figure 17 materials-12-00825-f017:**
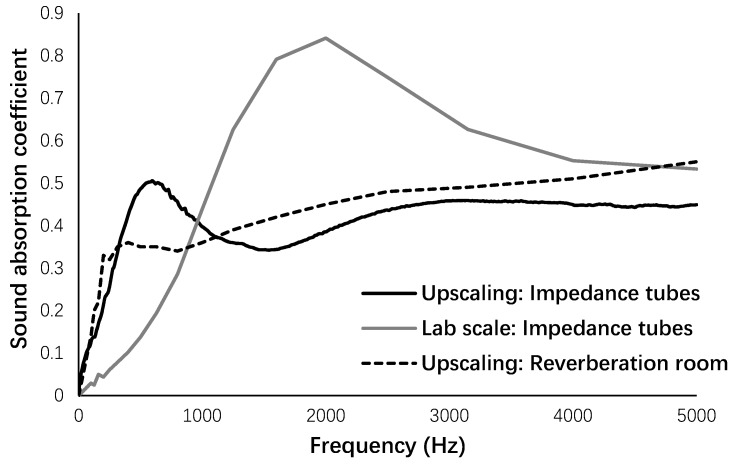
A comparative assessment of sound absorption.

**Figure 18 materials-12-00825-f018:**
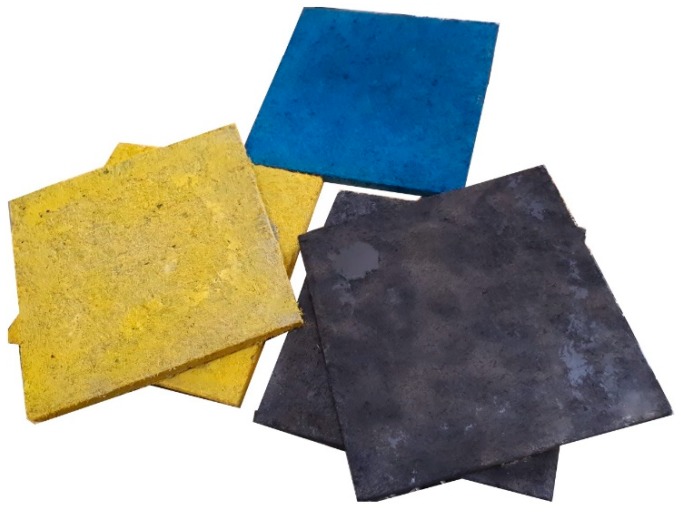
Colorful acoustic panels for assembling on internal walls.

**Table 1 materials-12-00825-t001:** Properties of alkali-activated foam concrete (by mechanical foaming method).

Ref.	Binder	Foaming			Alkali Activator	A^[a]^/B^[b]^	Density (Kg/m^3^)	Porosity (%) (Used Method)	Compressive Strength (MPa) (28 days)	Thermal Conductivity (W/m·K)	Sound Absorption	Porous Structure	Curing
		Method	Foaming Agent	Content								Pore Size (µm)	
[16]	GGBFS^[c]^	Mechanical	Foaming agent (in form of protein hydrolysate)	10–35% (wt.% of binder)	NaOH (12 M) and Na_2_SiO_3_ (Ms.^[d]^ = 2.5)	0.56	550–1500	45–65% (SEM^[e]^ and image analysis software “Fiji software”)	2.5–13	n.d.^[f]^	Acoustic absorption coefficients higher than 0.5 in the medium- to high-frequency regions	<100–1000	Sealed in plastic bags until the testing day (28 days) at room temperature (23 °C)
[17]	GGBFS/FA^[g]^	Mechanical	Protein based agent	62–76% (volume ratio)	Three types: 1—Ca(OH)_2_ and Mg(NO_3_)_2_ 2—Ca(OH)_2_ and Na2SiO_3_ 3—Ca(OH)_2_ and Na_2_SiO_3_	0.1–0.16	325–492	n.d.	0.5–2	0.088–0.121	n.d.	<0.01–100	Sealed and cured at room temperature until the testing day
[18]	FA/GBFS^[h]^	Mechanical	Commercial surfactant diluted with water	Until the required density achieved	Anhydrous sodium metasilicate (Solid activator—one part geopolymer)	0.085	600–1200	n.d.	(28 days) (1) with no aggregate = 1.6–13 (2) Sand as fine aggregate = 1–9 (3) Glass as fine aggregate = 1.75–7	(1) with no aggregate = 0.18–0.872 (2) Sand as fine aggregate = 0.18–0.705 (3) Glass as fine aggregate = 0.15–0.937	n.d.	n.d.	Sealed and cured at ambient temperature until the day of testing
[19]	GGBFS/FA	Mechanical and Chemical	(Mechanical foaming) SDS^[i]^ with and without foam stabilizer (Chemical foaming) H_2_O_2_ modified with (SDS) with and without foam stabilizer	N.A.^[j]^	NaOH (3 M) and Na_2_SiO_3_ (Ms. = 2)	0.38	650–660 (for mechanical foaming and chemical foaming)	n.d.	(Mechanical foaming) 1.75–2.25 (Chemical foaming) 1.6–1.9	0.2–0.27	n.d.	<5–1470	Sealed and cured at 60 °C for 24 h, then sealed at room temperature until the testing day
[20]	Class F-FA/GGBFS	Mechanical	Diluted aqueous surface-active concentrate	5–16% (wt.% of binder)	NaOH (12 M) and Na_2_SiO_3_ (Ms. = 2)	0.395	720–1600	n.d.	3–14	0.15–0.48	Acoustic absorption coefficients of 0.7–1.0 at 40–150 Hz, and 0.1–0.3 at 800–1600 Hz	n.d.	Sealed in a plastic bag and cured at 40 °C for 24 h, then 27 days in the ambient conditions
[21]	Class C-FA	Mechanical	NA.	foam: geopolymer paste = 2:1 (by volume)	NaOH (12 M) and Na_2_SiO_3_ (Ms. = 3.2)	0.5	1650 (room temperature) 1667 (60 °C)	15.29% (room temperature) 6.78% (60 °C) (SEM and image analysis software)	3.3–18.1 (room temperature) 11–18.2 (60 °C)	n.d.	n.d.	4–37	Room temperature or 60 °C for 24 h, then unsealed in the open air until the testing day
[13]	GBFS	Mechanical	SDS	11.5% (wt.% of binder)	NaOH and Na_2_SiO_3_ (Ms. = 3.2)	0.436–0.574	450–780	n.d.	0.2–0.48	0.0023	n.d.	<500–5500	Sealed and cured at 60 °C for 24 h, then kept sealed and cured in room temperature until testing day
[22]	Class F-FA/GGBFS	Mechanical	Synthetic organic foaming agent	3.3–16% (wt.% of binder)	NaOH (12 M) and Na_2_SiO_3_ (Ms. = 2)	0.395	n.d.	44–65% (1—water saturation. 2—SEM 3—image analysis (IA) system consisting of an Olympus optical microscope and the Analysis-FIVE software)	Compressive strength (90 days) (3.4 ± 0.7)–(16.2 ± 3.3)	n.d.	n.d.	0.01–100	Sealed and cured at 40 °C for 24 h, then aged at room temperature for 90 days
[23]	MK^[k]^/BFS	Mechanical	Aqueous solution of a foaming agent diluted with water	(31–43.6 kg/m^3^)	NaOH and Na_2_SiO_3_	0.6	400–1000	n.d.	0.4–11	n.d.	acoustic absorption coefficients of 0.15–0.9 at 100–4000 Hz	0.1–0.5 mm	Cured at room temperature for 28 days
[24]	CRS^[l]^ powders/BFS^[m]^	Mechanical	Aqueous solution of a foaming agent diluted with water	As the required density achieved	NaOH and Na_2_SiO_3_ (Ms. = 2.87)	0.4 and 0.5	700–1000	30–90% (Water absorption)	Compressive strength 2–5 Flexural strength 0.25–0.6	n.d.	Transmission Loss = 30–50 dB at 100–5000 Hz	0.1–1 mm	Cured in air until the testing day

A^[a]^: alkaline activator; B^[b]^: binder; GGBFS^[c]^: ground granulated blast-furnace slag; Ms.^[d]^: SiO_2_/Na_2_O or K_2_O; SEM^[e]^: scanning electron microscope; n.d.^[f]^: not detected; FA^[g]^: fly ash; GBFS^[h]^: granulated blast-furnace slag; SDS^[i]^: sodium dodecyl sulphate solution; N.A.^[j]^: not available; MK^[k]^: metakoalin; CRS^[l]^: calcined reservoir sludge; BFS^[m]^: blast-furnace slag.

**Table 2 materials-12-00825-t002:** The mix proportion of alkali-activated foam concrete (based on mass ratios). SS—sodium silicate; SH—sodium hydroxide; PVA—polyvinyl alcohol.

Slag	Recycled Aggregates/Slag	Alkali Activator/Slag	SS/SH	PVA Fiber/Slag	Foam/Slag
1	1	0.56	2.5	0.024	0.25

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
