# Peer review of "Impacts of Casting Scales and Harsh Conditions on the Thermal, Acoustic, and Mechanical Properties of Indoor Acoustic Panels Made with Fiber-Reinforced Alkali-Activated Slag Foam Concretes"

_materials, 2019, doi:10.3390/ma12050825_

Reviewer 1 Report

Very interesting paper. Please correct minor issues:

Figure 12) some of the images are not clear

Figure 13)  the legend should also include the graph in black

Figure 16 b) vertical axis and the legend should be corrected. Vertical axis should be moved out of the numbers, and the legend should include both lines

Author Response

First of all, the authors would like to thank the Reviewers for their thorough examination of the paper and valuable comments, which have contributed to improving the overall quality of the work.

Remark #1: Figure 12) some of the images are not clear
Reply: The authors agree with the Reviewer. However, to have very clear figures with using optical images, it is required to have a perfectly smooth surface of the fracture specimens. Providing a perfectly smooth surface using water jet or other devices damages to the porous structures of the materials and comparing the results become impossible. Therefore, the authors tried to show the highest resolution of the porous structure of materials under different conditions without damaging their internal structures.

Remark #2: Figure 13)  the legend should also include the graph in black
Figure 16 b) vertical axis and the legend should be corrected. Vertical axis should be moved out of the numbers, and the legend should include both lines

Reply: The authors agree with the Reviewer. The authors considered both comments in the original version of the paper. Since the paper was submitted to the journal as word files and the authors could not finalize the sent pdf version to the Reviewer, these mistakes have appeared in the received pdf version by the Reviewers. However, the authors replaced these figures again to avoid happening these problems in the pdf version received by the Reviewers.

Reviewer 2 Report

Several points have to be revised. 

1, Line 60. Grammar. Are or have been developed. Line 75-76. Grammar. Incomplete sentence.  

2, In the Introduction, the aims of the paper were put too dispersedly. The sentences in the last 2 paragraphs of Introduction need be more clearly organized. 

3, The discussion in 3.1 used the term “mass”, but Figure 7 showed “density”. Keep the terms constant, although they actually mean the same thing. 

4, The structure of section 3. The title of 3.1 and 3.2 are "effects of carbonation and freeze/thaw” but you only showed the effects on mass. In 3.3-3.5, the properties are all affected by the two conditions. Why are they not “effects of carbonation and freeze/thaw”? So the structure of section 3 needs to be better organized.  

5, The legend for the Ambient condition curve in Figure 9 (b) is missing. The same to many other figures. Please pay more attention on the result figures. 

6, Regarding the reaction products after carbonation, you mentioned several times "calcite" without experimental proof like XRD. This statement may not be a problem for OPC system. But for AAS system, there can really be Na2CO3 due to the activator you used. Although identifying the crystals may not influence the conclusions, it is still good to either prove that or state that less affirmatively.  

7, Considering the order you present the results, it is good to change the tile of the paper to “Multi....: Mechanical, acoustic, and thermal properties”?

Author Response

First of all, the authors would like to thank the Reviewers for their thorough examination of the paper and valuable comments, which have contributed to improving the overall quality of the work.

Remark #1: Line 60. Grammar. Are or have been developed. Line 75-76. Grammar. Incomplete sentence.

Reply: The authors agree with the Reviewer. Therefore, these sentences were revised in the text.

Remark #2: In the Introduction, the aims of the paper were put too dispersedly. The sentences in the last 2 paragraphs of Introduction need be more clearly organized.

Reply: The authors agree with the Reviewer. These two sentences were re-written in the text to provide clearer organization for readers.

Remark #3: The discussion in 3.1 used the term “mass”, but Figure 7 showed “density”. Keep the terms constant, although they actually mean the same thing.

Reply: The authors agree with the Reviewer. The term "mass" changed to "density". Thanks for your useful comment.

Remark #4: The structure of section 3. The title of 3.1 and 3.2 are "effects of carbonation and freeze/thaw” but you only showed the effects on mass. In 3.3-3.5, the properties are all affected by the two conditions. Why are they not “effects of carbonation and freeze/thaw”? So the structure of section 3 needs to be better organized.

Reply: The authors agree with the Reviewer. The structure of section 3 has been changed to have a better organization for readers to understand clearly.

Remark #5: The legend for the Ambient condition curve in Figure 9 (b) is missing. The same to many other figures. Please pay more attention on the result figures.

Reply: The authors agree with the Reviewer. The authors considered all these points in the original version of the paper. Since the paper was submitted to the journal as the word files and the authors could not finalize the sent pdf version to the Reviewer, these mistakes have appeared in the received pdf version by the Reviewers. However, the authors replaced these figures (also, figures 13 and 16b) again to avoid happening these problems in the pdf version received by the Reviewers.

Remark #6: Regarding the reaction products after carbonation, you mentioned several times "calcite" without experimental proof like XRD. This statement may not be a problem for OPC system. But for AAS system, there can really be Na2CO3 due to the activator you used. Although identifying the crystals may not influence the conclusions, it is still good to either prove that or state that less affirmatively.

Reply: The authors agree with the Reviewer. The authors changed "the crystalline calcite" to "the crystals". As the Reviewer mentioned that identifying the crystals may not influence the conclusions, however, the authors think the main crystals are included of calcite and sodium carbonate.

Remark #7: Considering the order you present the results, it is good to change the tile of the paper to “Multi....: Mechanical, acoustic, and thermal properties”?

Reply: The authors agree with the Reviewer. The title of the paper was changed to "Impacts of casting scales and harsh conditions on the thermal, acoustic, mechanical properties of indoor acoustic panels made with fiber-reinforced alkali-activated slag foam concretes".

Reviewer 3 Report

This study presents the experimental results regarding the development of fiber-reinforced alkali-activated slag foam concretes in upscaling. Mechanical, acoustic, and thermal properties of acoustic panels were assesse. The experimental work and the conclusions are very interesting. This work is very  good.

Author Response

The authors would like to thank the Reviewer for your thorough examination of the paper and valuable comment about our work.